



# Employing relaxed smoothness constraints on imaginary part of refractive index in AERONET aerosol retrieval algorithm.

Alexander Sinyuk[1,2], Brent N. Holben[2], Thomas F. Eck[3,2], David M. Giles[1,2], Ilya Slutsker[1,2], Oleg Dubovik[4], Joel S. Schafer[1,2], Alexander Smirnov[1,2], and Mikhail Sorokin[1,2].

[1]Science Systems and Applications, Inc. (SSAI), Lanham, MD 20706, USA
[2]NASA Goddard Space Flight Center (GSFC), Greenbelt, MD 20771, USA
[3]University of Maryland Baltimore County (UMBC), Baltimore, MD 21250, USA
[4]Univ. Lille, CNRS, UMR 8518 – LOA – Laboratoire d'Optique Atmospherique, Lille, France

*Correspondence to*: Alexander Sinyuk (aliaksandr.sinyuk-1@nasa.gov)

**Abstract**. In the Aerosol Robotic Network (AERONET) aerosol retrieval algorithm, smoothness constraints on the imaginary part of the refractive index (IPRI) provide control of retrieved spectral dependence of aerosol absorption by preventing the inversion code from fitting the noise in optical measurements and thus avoiding unrealistic oscillations of retrievals with wavelength. The history of implementation of the IPRI smoothness constraints in the AERONET aerosol retrieval algorithm is discussed. It is shown that the latest version of the IPRI smoothness constraints, termed standard (STD) and employed by Version 3 (V3) of aerosol retrieval algorithm, should be modified to account for strong variability of light absorption by brown carbon (BrC) containing aerosols in UV through mid-visible parts of the solar spectrum. In V3 strong spectral constraints were imposed at high values of the Angstrom Exponent (AE; 440-870 nm) since black carbon (BC) was assumed to be the primary absorber, while the constraints became increasingly relaxed as AE deceased to allow for wavelength dependence of absorption for dust aerosols. The new version of the IPRI smoothness constraints assigns different weights to different pairs of wavelengths which are the same for all values of the Angstrom Exponent. For example, in the case of four wavelength input, the weights assigned to short wavelength pairs (440-675, 675-870 nm) are small ($10^{-6}$) so that smoothness constraints do not suppress natural spectral variability of the IPRI. At longer wavelengths (870-1020 nm), however, the weight is ten times higher to provide additional constraints on the IPRI retrievals of aerosols with high AE due to low sensitivity to aerosol absorption for longer channels at relatively low aerosol optical depths for these fine mode dominated aerosols. The effect of applying the new version of the IPRI smoothness constraints, termed relaxed (REL), on retrievals of single scattering albedo (SSA) is analysed for case studies of different aerosol types: BC and BrC containing fine mode aerosols, mineral dust coarse mode aerosols and urban industrial fine mode aerosol. It is shown that for BrC containing aerosols employing the REL smoothness constraints resulted in significant reduction, compared to STD, in retrieved SSA and spectral residual errors at the short wavelengths. For example, biomass burning smoke cases show a reduction in SSA and spectral residual at 380 nm is ~0.033 and ~17% respectively for the Rexburg site and ~0.04 and ~ 12.7% for the Rimrock site, both AERONET sites in Idaho, USA. For a site with very high levels of BC containing aerosols (Mongu , Zambia) the effect of modification in the IPRI smoothness constraints is minor. For mineral dust aerosols at small



AE values (Mezaira site, UAE) the spectral constraint on IPRI was already relaxed in V3 therefore the new REL constraint results in minimal change. In the case of weakly absorbing urban industrial aerosols at the GSFC site, there are significant changes in retrieved SSA using the REL assumption, especially reductions at longer wavelengths: ~ 0.016 and ~0.02 at 875
and 1020 nm respectively for 440 nm AOD ~0.3. Comparison of aerosol parameters retrieved by inversion using STD and REL assumptions is presented. Analysis is done for retrievals utilizing the four standard AERONET wavelengths obtained at four AERONET sites: Rexburg, Mongu, Mezaira, and GSFC. The largest difference is found for the IPRI retrievals for BrC containing biomass burning (Rexburg) and urban industrial (GSFC) aerosols in which cases employing the STD assumption in the AERONET inversion was over constraining the spectral dependence of the IPRI. The modification of smoothness
constraints on the IPRI has a minor effect on retrievals of other aerosol parameters such as the real part of refractive index and parameters of the aerosol size distribution. Both SSA retrieved using STD and REL assumptions were compared to SSA derived from in situ measurements collected during the DRAGON-MD field campaign in 2011. The DISCOVER-AQ column-integrated in situ aircraft SSA data for 550 nm were compared to AERONET retrievals at 440 nm and 675 nm which were interpolated to 500 nm and showed a closer agreement between in situ SSA and SSA retrieved from inversions
employing the REL assumption than between in situ SSA and SSA retrieved using STD constraints. The implementation of the relaxed smoothness constraints on the imaginary part of the refractive in the next version of the AERONET inversion algorithm will produce significant impacts at some sites with changes up to ±0.033 and ±0.017 in short wavelength channels (380nm and 440nm) for some biomass burning smoke cases with significant BrC content and possibly up to ±0.015 in mid-visible channels (500nm and 675nm) to near IR channels (870nm to 1020nm) for some urban industrial aerosol types while
still mostly within the uncertainty of the AERONET SSA retrievals, depending on AOD level, Angstrom Exponent and wavelength. For mineral dust aerosols the impact will be insignificant, while for biomass burning aerosol dominated by BC the changes will be relatively small.

## 1. Introduction.


The Aerosol Robotic Network (AERONET) (Holben et al, 1998) of globally distributed ground-based sun-sky radiometers provides measurements of total atmospheric column spectral aerosol optical depth (AOD) and inversion algorithm retrievals of column integrated aerosol size distribution (ASD) and complex index of refraction (CIR). The AOD measurements have been frequently utilized for satellite validation purposes (Sayer et al., 2018, 2019; Levy et al., 2013,
2015; Holzer-Popp et al., 2013; Lyapustin et al., 2018; Kahn et al., 2010; Limbacher et al., 2019; Ahn et al., 2014; Choi et al., 2018) plus as input to the inversion algorithm along with directional sky radiances over a range of scattering angles. The AOD are measured at high accuracy (Eck et al., 1999; ~0.01 in visible and near infrared and ~0.02 in the UV) thereby providing a strong bound on the inversion results, especially since the sky radiance uncertainty combined with extra-terrestrial flux uncertainty is ~5% (Sinyuk et al., 2020). The retrieved aerosol parameters are often used in development of



satellite retrieval algorithms which must sometimes assume some aerosol optical and physical properties (Remer et al, 2005; Lyapustin et al., 2018). The retrievals of aerosol parameters are performed by the AERONET aerosol inversion algorithm which was developed by Dubovik and King (2000) and refined by Dubovik et al. (2006) with the addition of non-spherical (spheroidal) scattering. The latest version of the algorithm employed in AERONET Version 3 (V3) is described in detail in Sinyuk et al. (2020). The data quality assurance procedures and cloud screening of AOD in V3 are presented in Giles et al.

(2019). The standard AERONET aerosol retrieval product is obtained by inverting measurements taken at four standard wavelengths: 440, 675, 875 and 1020 nm. However, the V3 aerosol retrieval algorithm can invert extended sets of wavelengths including the UV at 380 nm (Sinyuk et al., 2020). Absorption at 380 nm is particularly important as this is the wavelength range that satellite observations and algorithms are able to retrieve atmospheric column absorption from existing (Jethva et al, 2014) and future satellite sensors (Werdell at al., 2019).  In addition to ASD parameters and spectral CIR which

are directly retrieved by the aerosol retrieval algorithm, other aerosol characteristics such SSA, absorption optical depth, asymmetry parameter, lidar and depolarization ratios are calculated from the  retrieved aerosol parameters.

In general, inverted measurements have limited information content resulting in non-uniqueness of the solution and high sensitivity to random measurement errors.  In the worst-case scenario, an inversion fits the measurements noise (overfitting of the data) causing non-realistic oscillation in retrieved functions (e. g. Twomey, 1977). To prevent overfitting

of the data, additional constraints on the retrieved functions should be imposed. These constraints usually restrict the norm of the solution or the norm of its derivatives (e. g. Dubovik, 2004, Aster et al., 2013) and are called smoothness constraints. The strength of the smoothness constraints should be selected in an optimal way as to prevent non-uniqueness of solution and overfitting of the measurements and yet not to over constrain and suppress the natural variability of retrieved functions.  In addition, stronger smoothness constraints should be optimally imposed at those ranges of solution variability which are not

sufficiently constrained by the measurements. For example, for high AE aerosols (fine mode dominated)  measurements sensitivity to the IPRI at longer wavelengths is limited due to low AOD at these wavelengths. The AERONET aerosol retrieval algorithm employs two types of smoothness constraints: (1) size dependence of ASD is constrained by restricting the norm of the third derivatives and (2) the spectral dependencies of the real and imaginary parts of refractive index are constrained by restricting that of the first derivatives (Dubovik and King, 2000).

The IPRI is the one of key retrieved aerosol parameters largely defining SSA which plays the central role in estimation of aerosol radiative forcing and atmospheric heating (Haywood and  Boucher, 2000; Jacobson, 2001; Bond et al., 2013; Myhre et al., 2013). The optimal selection of the IPRI smoothness constraints and its effect on retrieved aerosol parameters is the subject of this paper.  Throughout the development of the AERONET aerosol retrieval algorithm different implementations of the IPRI smoothness constraint were employed. The latest V3 implementation imposes weak smoothness

constraints on the spectral dependence of the IPRI for low AE aerosols and strong smoothness constrains for high AE aerosols with linear interpolation on AE between these two cases. The first assumption works reasonably well for coarse mode dominated mineral dust aerosols which exhibit high IPRI spectral variability in UV to mid-visible parts of solar spectrum and the second assumption is suitable for aerosols whose chemical composition is dominated by BC exhibiting



spectrally flat imaginary index of refraction (Bond and Bergstrom, 2006; Kirchstetter et al., 2004). However, for high AE
aerosols containing BrC, the assumption of the strong IPRI smoothness constraints is not always suitable due to possible
strong spectral variability of absorption by these aerosols at short wavelength (mid-visible through the UV).

This paper describes the modification of V3 assumptions on the IPRI smoothness constraints to accommodate a priori
information on spectral variability of the IPRI for different aerosol types. Section 2 discusses the history of implementation
of the IPRI smoothness constraints in the AERONET aerosol retrieval algorithm and presents the new implementation which
is termed relaxed smoothness constraints (REL). In Section 3 we describe the effect of the REL smoothness constraints on
SSA retrievals by analysing case studies for different aerosol types. Comparisons of aerosol parameters retrieved by the
AERONET aerosol retrieval algorithm using both REL and STD assumptions on the IPRI smoothness constraints are
presented in Section 4. Section 5 shows examples of comparison of both REL and STD SSA retrievals to SSA derived from
in situ measurements. The summary and conclusions are presented in Section 6.


## 2. Theory

The AERONET aerosol retrieval algorithm, like that of GRASP (Dubovik et al., 2011, Dubovik et al., 2021), is based
on the Multi-term Least Square Method (LMS) approach, which was developed over the years by O. Dubovik with co-
authors (Dubovik et al., 1995, Dubovik and King, 2000, Dubovik, 2004, Dubovik et al., 2011, Dubovik et al., 2021). The
Multi-term LSM concept allows flexible incorporation of very general a priori constraints on retrieved parameters,  with
emphasis on smoothness constraints in particular. Smoothness constraints limit variability of retrieved functions such as
aerosol size distribution and spectral dependence of refractive index by using a priori information on their derivatives. From
a formal point of view (e.g., Dubovik et al., 2021), the smoothness constraints are related to the limited values of derivatives
of retrieved functions, i. e. with deviations of their m-th derivatives from zero:

$$\frac{\partial^m f}{\partial x^m} \approx 0. \tag{1}$$

For the vector of unknows $\boldsymbol{a} = [a_1, a_2, ..., a_n]^T$, which is a discrete representation of continuous function $f$ , the condition
(1) can be formulated as a linear system of equations

$$\boldsymbol{G}_m \boldsymbol{a} + \Delta_g^* = \boldsymbol{0}^*, \tag{2}$$

where $\boldsymbol{G}_m$ is the Jacobian matrix of m-th derivatives which in the discrete case is approximated by the matrix of m-th finite
differences estimated at point $\boldsymbol{a}$. $\boldsymbol{0}^*$ is the zero vector, representing the fact that a priori estimates of the corresponding
derivatives are equal to zero. $\Delta_g^*$ are the errors reflecting uncertainty in the knowledge of the deviation of the retrieved
function $f$ from the assumed simple functions such as constant (m=1), straight line (m=2), parabola (m=3), etc. Under the
assumption that the $\Delta_g^*$ in (2) are normally distributed with covariance matrix $\boldsymbol{C}_g$, smoothness constraints can be easily
included in the general framework of Multi-term LMS technique to obtain the iterative solution for vector $\boldsymbol{a}$. For the case of


a standard set of AERONET observations (spectral AOD, spectral sky radiances) and retrieved aerosol parameters (ASD, CIR), the solution looks as follows:

$$\left(\sum_{k=1}^{2}\gamma_k \boldsymbol{K}_{k,p}^{T}\left(\boldsymbol{W}_k\right)^{-1}\boldsymbol{K}_{k,p} + \sum_{n=1}^{3}\gamma_n\boldsymbol{\Omega}_n\right)\Delta\boldsymbol{a}^{p} = \sum_{k=1}^{2}\gamma_k \boldsymbol{K}_{k,p}^{T}\left(\boldsymbol{W}_k\right)^{-1}\Delta\boldsymbol{f}_k^{p} + \sum_{n=1}^{3}\gamma_n\boldsymbol{\Omega}_n\boldsymbol{a}^{p}. \tag{3}$$

In Eq. (3) $\Delta\boldsymbol{f}_k^{p} = \boldsymbol{f}_k^{*} - \boldsymbol{f}_k^{p}$, where $\boldsymbol{f}_k^{*}$ are vectors of measurements of AOD (k=1) and sky radiances (k=2) and vectors $\boldsymbol{f}_k^{p}$ being the measurements fit at p-th iteration. $\boldsymbol{K}_{k,p}^{T}$ k=1,2 are Jacobian matrices at p iteration of the function $\boldsymbol{f}_k$ in the vicinity of $\boldsymbol{a}^{p}$ and $\boldsymbol{W}_k$ denotes weighting matrices related to corresponding covariance matrices $\boldsymbol{C}_k$ by

$$\boldsymbol{W}_k = \frac{1}{\varepsilon_k}\boldsymbol{C}_k, \tag{4}$$

where $\varepsilon_k$ is the first diagonal element of $\boldsymbol{C}_k$ and $\gamma_k = \frac{\varepsilon_1^2}{\varepsilon_k^2}$ are Lagrange multipliers. $\boldsymbol{\Omega}_n$ denotes smoothness matrices for size distribution (n=1), real (n=2) and imaginary parts (n=3) of the complex index of refraction and are defined as

$$\boldsymbol{\Omega}_n = \boldsymbol{G}_{m,n}^{T}\boldsymbol{W}_{m,n}^{-1}\boldsymbol{G}_{m,n}. \tag{5}$$

The values of Lagrange multipliers $\gamma_n$ in Eq. (3) determine the strength of the corresponding smoothness constraints. Their selection is described in detail in (Dubovik and King, 2000, Dubovik, 2004), where it is shown that theoretical values of the Lagrange multiplier can be expressed as a function of the norm of m-th derivatives. In practice, however, it was found both convenient and justifiable to determine the values of the Lagrange multiplier for the smoothness constraints empirically based on external information on the variability of the retrieved functions. For example, if external information suggests weak variability of refractive index with wavelength, the value of Lagrange multiplier can be adjusted as to force spectral dependence of refractive index to be spectrally flat. From other hand, if substantial spectral variability is expected, the strength of smoothness constraint should be relaxed.

The primary goal of smoothness constraints on the IPRI is to provide control of retrieved spectral dependence of aerosol absorption by preventing excessive destabilising effect of measurement noise on the solution and thus avoiding unrealistic oscillations of retrievals with wavelength. Also, they help in stabilizing retrievals of absorption of radiation for high AE aerosols at long wavelengths by constraining its spectral dependence based on a priori information. The long wavelengths in these high AE cases often have AOD that is too low to contain sufficient information content on particle absorption.

The AERONET aerosol retrieval algorithm employs constraints of the first derivatives of the IPRI wavelength dependence (m=1) in which case these constraints relate the values of the IPRI at neighboring pairs of wavelengths. Throughout the course of algorithm development, different approximations were used for Lagrange multipliers $\gamma_3$ while assuming equal weights to all the pairs of wavelengths and thus replacing weighing matrices $\boldsymbol{W}_{m,n}$ with unit matrices. In version one (V1) of the AERONET database, it was assumed that the IPRI for all the aerosol types is spectrally flat. This was physically justified by the flat spectral dependence of the principal fine mode aerosol absorber, black carbon (BC), in this wavelength range. Therefore, the strong smoothness constrains were used with the value of $\gamma_3$ equal to $10^{-1}$. In the



AERONET version two and three (V2 and V3) it was realized that such strong smoothness constraints suppress the natural
spectral variability of absorption of dust aerosols (low AE), especially for the short wavelength visible (440 nm), but still
assumed flat spectral independence of the IPRI for fine mode aerosols. The Lagrange parameter $\gamma_3$ was linearly
interpolated by AE (440-870 nm) between "pure" dust (AE=0.001, $\gamma_3 = 10^{-6}$ ) and "pure" fine mode (AE=2.5, $\gamma_3 = 10^{-1}$ ).

After incorporating the vector radiative transfer model SORD (Korkin et al., 2017) in the V3 aerosol retrieval algorithm
(Sinyuk et al., 2020) it became possible to invert an extended set of wavelengths including in the near UV (380 nm). In this
case however, the assumption on spectral independence of the IPRI for fine mode aerosols might not be very realistic for the
high AE aerosols containing BrC with strong UV absorption (e. g. Mok et al., 2016). Therefore, the V3 assumption for the
IPRI smoothness constraints should be modified by relaxing its strength at short wavelength but keeping larger $\gamma_3$ value at
longer wavelength where measurement signal is low, thus making strength of the smoothness constraints spectrally
dependent. This can be accomplished by adjusting the elements of weight matrix in Eq. (5) by using smaller weights for
short wavelength pairs and the larger ones for those of longer wavelengths.

In the new assumption of the IPRI smoothness constraints, which we term "relaxed smoothness constraints" (REL), the
value of $\gamma_3$ in Eq. (3) is kept the same as for the "pure" dust in V2 and V3 ( $10^{-6}$ ). The diagonal elements of weighting
matrix are equal to one for wavelength pairs, which, for example, in the case of six wavelength input are 380-440, 440-500,
500-675, 675-870 nm. However, for 870-1020 nm pair, the diagonal element of weighting matrix is equal to 10 thus making
effective value of $\gamma_3 = 10^{-5}$. The performance of this assumption is tested for different aerosol types in the following
sections.

### 3.   Effect of relaxed smoothness constraints for the IPRI on SSA retrievals.

In this section, the effect of incorporating the relaxed smoothness constraint in the AERONET aerosol retrieval
algorithm on SSA retrievals is analysed for different aerosol types: brown and black carbon containing aerosols, desert dust,
and urban industrial aerosols.

3.1 Brown and black carbon containing aerosols.
The light absorbing aerosols that are produced from combustion (e. g. from biomass burning and fossil fuels) are
typically classified as black carbon or brown carbon (e. g. Adler et al., 2019, Bond et al, 2006). BC absorbs radiation across
the entire  UV- near infrared spectrum exhibiting little to no spectral selectivity (the IPRI is spectrally flat). BrC possess a
strong wavelength dependent absorption that peaks in the UV spectral region and declines though the visible part of the
spectrum. (e. g. Kirchstetter and Thatcher, 2012).
Figure 1 shows two examples of SSA retrieved from six spectral channel measurements at 380, 440, 500, 675, 870 and
1020 nm taken at AERONET sites where BrC containing aerosols from biomass burning were present during the time of
observations: Rexburg, Idaho, USA on August 6, 2017 (Fig. 1a) and Rimrock, Idaho, USA August 23, 2018 (Fig. 1b). The



AE was high for both of these cases, AE (440-870 nm)=1.74 for the Rexburg case and AE= 1.79 for the Rimrock retrieval
case thereby imposing strong constraints on spectral IPRI in the V3 or STD retrievals. It is noted that the AOD was very
high for both of these cases (Rimrock: 1.65 at 440 nm and 0.35 1020 nm; Rexburg: 1.32 at 440 nm and 0.27 at 1020 nm)
thereby providing large aerosol signal at most wavelengths thus enabling very accurate retrievals of absorption information
(Sinyuk et al, 2020). Sky radiances at 440 to 1020 nm for these measurements were calibrated using an integrating sphere (e.
g. Holben et al., 1998) while the vicarious method (e. g. Li et al., 2008) was used for calibration at 380 nm since the sphere
does not provide high enough radiance output in the UV. Figure 1 displays two types of SSA retrievals: first, depicted as
STD, was obtained by inversion using the V3 assumption on the IPRI smoothness constraints, the second, depicted as REL,
is the result of inversion employing the new assumption on spectral IPRI while all other aspects remain the same. Two major
features are displayed in Fig.1: a significant decrease in SSA retrieved using REL at the shortest wavelengths (380 and 440
nm) and a reduction in corresponding residual error values at these channels. The residual is the root mean square difference
between the measured sky radiances and those computed based on the retrieved aerosol parameters. For, example, the
reduction in SSA in the Rexburg case is ~ 0.033 and ~0.013 at 380 and 440 nm respectively while at longer wavelengths, the
difference in SSA retrieved using REL and STD assumptions is less than 0.01. The spectral dependence of SSA retrieved
using REL assumption is qualitatively consistent with that of BrC absorption. The absolute reduction (difference between the
STD and the REL residual errors) in residual values in the Rexburg case is ~ 17% and ~5% at 380 and 440 nm respectively
implying more accurate SSA retrievals obtained using REL assumption. The retrieved spectral dependence of SSA and
residuals in the Rimrock case exhibit similar behavior: ~ 0.04 and ~ 0.012 decrease in SSA at 380 and 440 nm respectively
with correspondent reductions in spectral residuals of ~ 11.1% and ~ 2.55%. Figure 1 shows that in both cases the SSA
absolute differences at 675 nm are comparable to that at 440 nm, which only partly can be explained by BrC absorption due
to its decrease with wavelength. Another possible reason for that is the larger uncertainty at 675 nm than at other
wavelengths in extra-terrestrial solar spectrum as discussed in (Sinyuk et al., 2020).
Figure 2 shows SSA retrieved at the Mongu Inn AERONET site in Zambia for cases with two different values of AOD
measured at 440 nm: 0.49 (Fig. 2a) and 0.89 (Fig. 2b). The AE is high for both of these cases (1.85 and 1.97 respectively)
therefore resulting in strong constraints in the spectral IPRI in the STD retrievals. The light absorption by aerosol at this site
is dominated by BC due to savanna burning with significant flaming phase combustion production of BC (Ward et al., 1996),
therefore inversions using STD and REL assumptions should result in similar retrievals, assuming BC dominates over BrC
absorption. This is indeed the case as can be seen from Figure 2. For the lower AOD case, the SSA and residual values
corresponding to both STD and REL assumptions are very close with SSA absolute difference below 0.0033 and that of
residual less than 0.14%. For the higher AOD case, the spectral dependencies show similar behaviour except for 380 nm,
where the difference in SSA and residual values is ~ 0.017 and 3.6% respectively. This increase in aerosol absorption at 380
nm can be explained by the presence of BrC in addition to BC in aerosol composition for this biomass burning event. Indeed
Kirchstetter et al. (2004) measured a significant BrC absorption signature in biomass burning smoke from savanna burning
in southern Africa.





The above example illustrates that for biomass burning aerosols dominated by BC, the BrC also may be present in aerosol composition with its relative concentration possibly increasing with AOD magnitude. To check this assumption, statistics of retrieved SSA and spectral residuals at the Mongu Inn site were generated by averaging them over narrow AOD bins. Figure 3 shows averaged SSA and spectral residual values for two bins in AOD: 0.4 – 0.43 (Fig. 3a) and 1.0-1.4 (Fig. 3b). As can be seen, for the lower AOD bin, the average SSA retrieved with the REL assumption does not show any indication of BrC carbon presence and is very close to that of STD retrieval with maximum difference of ~ 0.009 at 675 nm. This difference, as before, can be attributed in part to larger uncertainty in solar spectrum at this wavelength. Spectral residual errors are very close for both types of inversion with the difference under 1%. Figure 3b, on the other hand, shows a slight increase in aerosol absorption at 380 nm as retrieved by inversion using the REL assumption with differences of ~0.006 and ~ 1.2% for SSA and residual respectively. The same type of statistics was generated for the Rimrock site and are shown on Figure 4. It shows averaged SSA and spectral residual values for two AOD bins: 0.5-0.53 (Fig, 4a) and 1.0-1.4 (Fig. 4b). A higher AOD magnitude bin for lower AOD for this site was selected because statistics for the 0.4-0.43 bin was not representative due to small sample size. Figure 4 shows that biomass burning at Rimrock shows strong BrC absorption at shorter wavelengths for AOD magnitude higher than 0.5 while, according to Figure 3b, for Mongu Inn BrC absorption at 380 nm is noticeable only for AOD magnitude higher than one ( an analysis for the 0.5-0.53 AOD bin for Mongu did not show any increase in 380 nm absorption). The magnitude of BrC absorption and its's strength relative to BC absorption (from mid-visible to UV) varies significantly for biomass burning aerosols potentially dependent on several factors including fuel types and moisture content, relative strengths of the phase of combustion (flaming versus smoldering), fire intensity, and ageing processes of the aerosols (Lewis et al., 2008; Di Lorenzo et al., 2017; Wong et al., 2019).

3.2 Desert dust aerosols.

For desert dust aerosols REL and STD retrievals of SSA are expected to be similar due to similarity between REL and V3 (or STD) assumptions and very weak constraints on spectral IPRI for low AE in both. Figure 5 shows SSA, and spectral residual values retrieved at the Mezaira, UAE AERONET site for two individual cases with of AOD: 0.44 (Fig. 5a) and 1.45 (Fig. 5b). The AE(440-870) was 0.24 for the case shown in Fig 5a and 0.20 for the case in Fig. 5b, therefore both are coarse mode dust dominated cases. As expected, SSA retrieved using REL and STD assumptions are very similar with differences below 0.002 and 0.004 for AOD 0.44 and 1.45 respectively. The difference between spectral residuals is within half a percent. The decrease in SSA at the shorter visible and UV wavelengths is primarily due to absorption from iron oxide content in mineral dust (Di Biagio et al., 2019). The similarity between SSA retrieved using REL and STD assumptions also holds for SSA averaged over AOD bins. Figure 6 shows averaged SSA and spectral residuals for two AOD bins 0.4-0.43 (Fig. 6a) and 1.0-1.4 (Fig. 6b). The SSA differences are below 0.0006 and 0.0015, therefore insignificant, for lower and higher AOD respectively with the differences in spectral residual values within 0.5 percent.

3.3 Urban industrial aerosols.



Urban industrial aerosols are defined as aerosol originating primarily from fossil fuel combustion in populated industrial regions (Eck et al., 1999). For this analysis the GSFC, Maryland, USA AERONET site located near to Washington DC, was selected. Note that this site has relatively high SSA when compared to other urban industrial sites (Dubovik et al. 2002; Giles et al., 2012), due to relatively low BC content. Figure 7 shows two cases of spectral SSA retrievals corresponding to two

different values of 440 nm AOD: 0.33 (Fig. 7a) and 0.49 (Fig. 7b). The AE(440-870 nm) for the case in 7a was 1.74 and for 7b was 1.63, therefore these high AE values resulted in strong constraint on IPRI being applied in the V3 or STD assumption. The AOD cases with lower magnitudes were selected for this site due to the lower average 440 nm AOD level at GSFC (0.186) as compared to the considered AERONET sites (e. g. 0.427 for Mongu Inn and 0.36 for Mezaira). In addition, according to Sinyuk et al., 2020, average SSA uncertainties at 440 nm estimated at GSFC for AOD (440)= 0.3 is ~ 0.03

which is similar to the AERONET threshold of AOD(440)=0.4 with SSA(440) uncertainty of 0.03 for Level 2. Figure 7 shows that employing the REL smoothness constraint assumption for IPRI reduces the values of spectral residuals in both AOD cases implying that the STD constraint was restricting spectral variability of the IPRI (SSA) for this aerosol type. The absolute SSA differences at the four standard AERONET sky radiance measurement channels, 440, 675, 870 and 1020 nm, for the first case are 0.0045, 0.0074, 0.0163, and 0.021 respectively, which is within average uncertainties estimated in

(Sinyuk et al., 2020) for this site at AOD 0.3 using STD assumption: 0.028, 0.034, 0.043, and 0.048. For the higher AOD case the SSA differences are smaller than that for lower AOD and within the average uncertainties of (Sinyuk et al., 2020) (in parentheses) estimated at AOD 0.5: 0.0074 (0.019), 0.0162 (0.023), 0.0154 (0.029), 0.0135 (0.033). The SSA differences at 380 nm are slightly larger than those at 440 nm: 0.0052 and 0.013 for Figure 7a and Figure 7b respectively. The SSA difference at 500 nm is in between of those at 440 and 675 nm.

Figure 8 shows spectral dependencies of retrieved SSA at the GSFC site averaged over narrow AOD bins: 0.3-0.33 (Fig. 8a) and 0.5-0.53 (Fig 8 b). As in the example cases analysed before, one can see a reduction in spectral residuals for the REL assumption and similar spectral behaviour of retrieved SSA. Also, both Figures 7 and 8 demonstrate similar features in spectral behaviour of SSA retrieved using the REL assumption, one of which is a slight drop in SSA values at 500 nm. The relative magnitude of this drop depends on SSA value at 675 nm which was largely suppressed by the STD assumption

(larger residual value at 675 nm) while at 500 nm both STD and REL residuals are lower than at neighbouring 675 nm and similar to each other. One of the possible reasons for this non-smooth spectral behaviour of retrieved SSA can be explained by spectrally non-uniform (in both magnitude and sign) sky radiance calibration coefficients which will have stronger effect on retrievals for weaker smoothness constraints. Figure 9 shows results of Figure 7a computed with SSA retrieved by inverting sky radiances which were vicariously calibrated using the technique of (Li et al., 2008). In this vicarious method

the FOV at 870 nm was computed from sphere radiance calibration along with the highly accurate sun calibration plus extra-terrestrial solar irradiance, and then subsequently utilized to determine sky radiance calibration for all other wavelengths by assuming a constant FOV for all wavelengths. Implemented in such a way, the vicarious calibration approach can potentially eliminate spectral non-uniformity of calibration as well as also eliminate dependence on solar spectrum at wavelengths other than 870 nm.  Figure 9 shows that inverting vicariously calibrated sky radiances produced smoother spectral dependence of





SSA eliminating the slight decrease of  SSA at 500 nm. Notably, all of the SSA plots for the GSFC site (Figs 7, 8 & 9) show a marked difference in spectral slope from less than 675 nm to greater than 675 nm. This is likely a result of BrC absorption in the shorter wavelengths from the organic carbon aerosol component present in emissions from fossil fuel combustion.

**4.    Comparison of aerosol parameters retrieved using STD and REL assumptions from four channel inversion.**


The comparison of retrievals obtained using REL vs STD assumptions on the IPRI smoothness constraints are presented for the Rexburg, Mongu, Mezaira and GSFC AERONET sites, and results are summarized in Tables 1 through 3. For parameters characterizing aerosol absorption, SSA comparisons are presented instead of IPRI due to its widespread use in climate research. In a manner similar to (Sinyuk et al, 2020), for SSA and the real part of refractive index (RPRI), this

analysis is done for three bins in 440 nm AOD which provides comprehensive comparisons by considering different levels of sensitivity to adjustment in the IPRI smoothness constraints. For ASD parameters, two bins in 440 nm AOD are used due to much higher stability and the small uncertainty of aerosol size distribution retrievals. All the Tables display mean values of the difference between STD and REL aerosol retrievals.

For Rexburg, the largest difference in SSA retrievals, shown in Table1, is at 440 nm with no clear dependence on the

level of AOD at 440 nm. The average difference (over AOD levels) of -0.0167 constitutes ~ 50 % of the SSA uncertainty at 440 nm estimated in Sinyuk et al., (2020) for biomass burning aerosols (for a different site however, Mongu). The SSA differences decreased as wavelength increased with the value of ~0.01 at 675 nm and the values at 870 and 1020 nm below 0.01 which is consistent with the spectral behaviour of BrC absorption. The differences in the retrieved real part of refractive index are shown in Table 2. Since assumptions on smoothness constraints for RPRI were not modified, the observed

differences are small, significantly smaller than variability in PRRI retrievals. Table 3 shows that retrieved ASD parameters for fine mode are practically the same for STD and REL inversions while the differences for volume median radius (VMR) for coarse mode are 0.014 μm  and 0.01 μm  for 440 nm AOD smaller and larger than 0.2 respectively. These differences in VMR for coarse mode are not significant considering a rather small coarse mode optical contribution for these fine mode dominated total AOD, plus the large size of these coarse mode particles (VMR typically >3 micron).

For Mongu, the differences in SSA are very small (Table 1) as expected due to weak spectral dependence of the IPRI for BC dominated aerosols, and this is the assumption applied in the STD retrievals. SSA differences at 440 nm: smaller than 0.0001 for first two 440 nm AOD bins. At the same time, the difference for 440 nm AOD greater than 0.6 is larger with the value of ~0.003 which may be indicative of BrC presence for this larger aerosol loading. Tables 2 and 3 show the mean difference in RPRI and ASD parameters respectively. As in the Rexburg case, the difference in RPRI retrievals is very small:

0.01 and lower. The retrieved ASD parameters are also in very close agreement  for STD and REL retrievals for both fine and coarse modes.

For Mezaira (UAE), the results of the REL-STD comparison are shown in Tables 1, 2 and 3. Due to similarity of REL and STD smoothness constraint assumptions for coarse mode aerosols, all three Tables show very small and insignificant



differences in the retrieved aerosol parameters. Table 1 shows that difference in SSA retrievals is smaller than 0.0001 except
at 675 nm where the difference in smaller than 0.002. The differences in RPRI retrievals are below 0.005 for all the wavelengths AOD levels. The difference in retrieved ASD parameters is also small with the largest difference (0.02 μm) in coarse VMR for AOD greater than 0.2, which is significantly smaller that standard deviation of retrievals for this parameter for this site (0.36 μm ).

Table 1 shows the differences in SSA retrievals for the GSFC AERONET site. The main feature as is displayed in Table
1 is the increase of differences with increasing AOD level at all wavelengths. If the increase is defined as the difference between average SSA values corresponding to third (>0.6) and second (0.4-0.6) AOD bins, the values are the following: 0.008, 0.005, 0.007 and 0.011 for 440, 675, 870 and 1020 nm respectively. This increase in REL-STD differences at higher AOD can be explained by stronger smoothing of the spectral dependence of IPRI by the STD constraints for larger AOD. This is illustrated by the different degree of reduction in spectral residual errors for different AOD values at 440 nm. For
example, Figure 7 shows that reductions in spectral residuals at 380 nm are 0.68 and 2.07% for AOD=0.33 and 0.49 respectively illustrating stronger suppression of spectral dependence of IPRI at higher AOD. Also, like in the case studies, the 675 comparison exhibits the largest difference which can be at least partially explained by the largest uncertainty in extra-terrestrial solar flux at this wavelength (Sinyuk et al., 2020) as well as possibly by spectrally non-uniform uncertainty in sky radiance calibration coefficients. Table 2 shows comparison of RPRI retrievals. The agreement between STD and
REL retrievals is very good with mean difference below 0.003. Table 3 also demonstrates a very good agreement in retrieved ASD parameters for this site, similar to all other sites examined .

### 5.  Comparison of SSA retrieved using STD and REL assumptions to in situ measurements.

In this section SSA retrieved from AERONET observations are compared to SSA determined from in situ measurements collected during the DRAGON-MD (Distributed Regional Aerosol Gridded Observational Network-Maryland) experiment in 2011 (Holben et al., 2018).  The SSA values are derived from in situ measurements made during aircraft vertical profiles of scattering and absorption coefficients at 550 nm using techniques described in more detail in Schafer et al., (2014).  The analysis of Schafer et al., (2014) had shown that SSA retrieved by AERONET (interpolated to 550 nm) were on average
0.011 lower than the values derived from in situ profiles. In this section we compare SSA derived from AERONET observations using both the STD and REL assumptions on the IPRI smoothness constraints to SSA determined from in situ measurements. The comparison is done for several temporary AERONET sites which were set up during the DRAGON 2011 campaign in Maryland, USA: Aldino, Beltsville, Essex, and Fair Hill sites. The type of aerosols that dominated the selected sites are similar to that at GSFC and can also be defined as urban industrial. All the comparisons are made for 440
nm AOD larger than 0.3 due to higher sensitivity to aerosol absorption for this aerosol type (Sinyuk et al., 2020).

Figure 10 shows comparisons of SSA retrieved at the Aldino site for three levels of 440 nm AOD: 0.352, 0.543, and 0.7. All three cases show that the values of SSA retrieved by the V3 inversion code employing STD assumption on the IPRI





smoothness constraints are lower than SSA derived from in situ measurements, which is consistent with the conclusion of Schafer et al., (2014). On the other hand, the SSA values retrieved from the inversion code employing the REL constraint

assumption are closer to in situ measured SSA. For the lowest 440 nm AOD value, Figure 10 a, the differences between in situ SSA and those retrieved using STD and REL (in parentheses) assumptions are 0.0158 (0.0024). For intermediate, Figure 10 b, and the largest, Figure 10 c, 440 nm AOD the corresponding deviations are: 0.0035 (-0.0027) and 0.011 (-0.001).

Figure 11 shows SSA comparisons for three AERONET sites: Beltsville (440 nm AOD=0.322), Essex (440 nm AOD =0.414, and Fair Hill (440 nm AOD=0.795). As in the Aldino cases, the SSA retrieved from inversions using REL

assumptions on IPRI smoothness constraints are closer to SSA derived from in situ measurements than those retrieved by the V3(STD assumption)  inversion code: 0.0166 (0.004) for Beltsville, 0.032 (0.018) for Essex, and 0.009 (0.0008) for Fair Hill.

The above examples demonstrate that the bias in SSA comparison reported in (Schafer et al., 2014), may be due at least in part to the strong IPRI smoothness constraints which restricted spectral variability of IPRI for this urban industrial aerosol

type.  Therefore, it is expected that analysis based on SSA retrievals from inversions employing the REL assumption on the IPRI smoothness constraints will results in smaller bias. However, it must be considered that these differences are well within the uncertainty error bars of both the in situ measured SSA and the AERONET retrieved SSA (regardless of the constraint type on the IPRI). Therefore the 'improvement' in agreement when applying the REL constraint on the IPRI may not have much significance.


## 6.    Summary and conclusions.

A modification of the assumption on the smoothness constraints of the spectral variation of IPRI employed by V3 of the

AERONET aerosol retrieval algorithm is presented and discussed. This modification is termed relaxed due to the weaker strength of this new smoothness constraint. It prevents over smoothing of the spectral dependence of the IPRI for different aerosol types including high AE aerosols (fine mode dominated) containing BrC. The modification employs spectrally dependent smoothness constraints which are implemented by assigning different weights to different pairs of wavelengths. The weight $10^{-6}$ is assigned to shorter wavelengths, while weight  $10^{-5}$ is assigned to the 870 – 1020 nm pair. This larger

weight provides stronger constraint for longer wavelengths where sensitivity to aerosol absorption is limited for high AE aerosols due to low AOD.

The effect of these REL smoothness constraints on retrievals of SSA was analysed for different aerosol types: BrC and BC containing aerosols, mineral dust, and urban industrial aerosols. Analyses have shown that the modification of the IPRI smoothness constraints mainly affect SSA retrieved for BrC containing biomass burning aerosols resulting in reduction of

both SSA and spectral residuals at shorter wavelengths (mid-visible to the UV). For example, for SSA retrieved for specific cases at the Rexburg and Rimrock AERONET sites(in Idaho) for biomass burning aerosol, the reduction in SSA and residual


at 380 nm are 0.033 and ~17% and 0.04, ~11.1% respectively. The reduction in sky radiance residual for these cases implies more accurate SSA retrievals with the new REL assumption since the computed sky radiances closely match the measured values. For mineral dust and BC dominated aerosols the effect of modified assumptions on IPRI smoothness constraints is

very small except for high AOD cases for BC dominated aerosols which show a slight increase in absorption at 380 nm. This can be explained by an increase of BrC in aerosol composition for high AOD biomass burning events dominated by BC as the primary aerosol absorber (Mongu, Zambia). The REL assumption on the IPRI smoothness constraints results in changes of SSA and spectral residual for urban industrial aerosols, especially reduced SSA at the longest wavelengths. For example, the reduction in retrieved SSA is ~ 0.016 and ~0.02 at 875 and 1020 nm respectively for 440 nm AOD ~0.3 at GSFC.

For urban industrial aerosols at GSFC, retrieved SSA exhibited non smooth spectral behaviour in the vicinity  of 500 nm. It was assumed that this behaviour can be explained, in part, by the spectral non-uniformity (in both magnitude and sign) of sky radiance calibration coefficients and spectral variation in uncertainty of the assumed extra-terrestrial irradiance. To check this assumption, SSA spectral dependence was retrieved by inverting vicariously calibrated sky radiances. Vicarious calibration employs field of view determined empirically by matching vicarious and sphere calibrated radiances at 870 nm

spectral channel.  Designed in such a way, it minimizes spectral non-uniformity of calibration as well as dependence on solar spectrum at wavelengths other than 870 nm. Inversion of the vicariously calibrated sky radiances produced smoother spectral dependence of SSA without an anomaly at 500 nm.

The average differences between aerosol parameters retrieved using STD and REL assumptions are presented and analyzed for four AERONET sites: Rexburg, Mongu, Mezaira, and GSFC. The analysis confirmed the results of the case

study analysis for SSA: the largest difference is observed for BrC containing biomass burning aerosols (Rexburg) and urban industrial aerosols (GSFC). The average differences for SSA retrieved at these sites are within U27 uncertainties estimated in (Sinyuk et al., 2020). For mineral dust (Mezira) and BC containing aerosols (Mongu) the average SSA differences are small due to similarity in STD and REL assumptions for dust cases and due to similarity in BC spectral variation of IPRI to the STD assumption for the high BC content cases. Comparisons of the RPRI and ASD parameters showed very close

agreements because assumptions on smoothness constraints for these parameters were not modified.

SSA retrieved from AERONET observations using both STD and REL assumptions were compared to SSA determined from in situ measurements collected from aircraft profiles during the DRAGON-MD experiment. The comparisons were done for four temporary AERONET sites set up during the experiment and showed closer agreement between SSA retrieved using the REL assumption  and in situ SSA than between in situ and SSA retrieved using the STD constraint. This can

possibly be explained by the fact that the STD assumption employed in V3 resulted in over-smoothing of the retrieved IPRI. The implementation of the relaxed smoothness constraints on the imaginary part of the refractive in the next version of the AERONET inversion algorithm will produce significant impacts at some sites with changes up to ±0.033 and ±0.013  in short wavelength channels (380nm and 440nm) for some biomass burning smoke cases with significant BrC content and possibly up to ±0.015 in mid-visible channels (500nm and 675nm) to near IR channels (870nm to 1020nm) for some urban

industrial aerosol types while still mostly within the uncertainty of the AERONET SSA retrievals, depending on AOD level,



Angstrom Exponent and wavelength. For mineral dust aerosols the impact will be insignificant, while for biomass burning aerosol dominated by BC the changes will be relatively small. The application of this new REL constraint on the spectral variation of the IPRI is being planned for future retrievals in Version 4 of the AERONET database. These retrievals will therefore have increased sensitivity to absorption by BrC in fine mode dominated aerosol than is currently available in the

Version 3 database. Additionally, future retrievals with the added wavelengths of 380 and 500 nm will enable more accurate and robust monitoring of BrC absorption in fine mode aerosols and iron oxides in coarse mode dust aerosols.

Author contributions. The development of relaxed smoothness constraints for AERONET aerosol retrieval algorithm is the result of joint effort of the members of AERONET team (listed as AS, BH, TE, DG, IS, JS, ASm, and MS) as well as

researchers from outside the project (OD). Individual contributions can be summarized as follows. AS modified AERONET retrieval code by incorporating relaxed smoothness along with OD who developed the theoretical basis. BH, TE, DG, IS, JS, ASm, and MS contributed to discussions during AERONET team meetings.

Competing interests. The authors declare that they have no conflict of interest.


Acknowledgements. The AERONET project at NASA GSFC is supported by the Earth Observing System Project Science Office cal–val, Radiation Sciences Program at NASA headquarters, and various field campaigns. Resources supporting this work were provided by the NASA High-End Computing (HEC) Program through the NASA Center for Climate Simulation (NCCS) at Goddard Space Flight Center.

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






| AOD (440 nm) | SSA (440 nm) | SSA (675 nm) | SSA (870 nm) | SSA (1020 nm) | Number of retrievals |
|---|---|---|---|---|---|
| Rexburg | | | | | |
| > 0.4 | -0.017 | 0.010 | 0.003 | 0.004 | 42 |
| 0.4-0.6 | -0.015 | 0.012 | 0.007 | 0.008 | 18 |
| > 0.6 | -0.018 | 0.010 | 0.000 | 0.000 | 24 |
| Mongu | | | | | |
| > 0.4 | 0.000 | 0.005 | 0.000 | 0.000 | 439 |
| 0.4-0.6 | 0.000 | 0.005 | 0.007 | 0.000 | 281 |
| > 0.6 | -0.003 | 0.005 | 0.001 | 0.001 | 158 |
| Mezaira | | | | | |
| > 0.4 | 0.000 | 0.001 | 0.000 | 0.000 | 1003 |
| 0.4-0.6 | 0.000 | 0.000 | 0.000 | 0.000 | 389 |
| > 0.6 | 0.000 | 0.002 | 0.000 | 0.000 | 614 |
| GSFC | | | | | |
| > 0.4 | 0.006 | 0.012 | -0.007 | -0.010 | 791 |
| 0.4-0.6 | 0.001 | 0.009 | -0.003 | -0.004 | 348 |
| > 0.6 | 0.009 | 0.014 | -0.010 | -0.015 | 443 |

Table 1. Average values, of the difference in SSA retrievals from inversions using STD and REL assumptions on IPRI smoothness constraints. The difference is defined as REL-STD.


| AOD (440 nm) | RPRI (440 nm) | RPRI (675 nm) | RPRI (870 nm) | RPRI (1020 nm) | Number of retrievals |
|---|---|---|---|---|---|
| Rexburg | | | | | |
| > 0.4 | 0.004 | 0.005 | 0.004 | 0.003 | 42 |
| 0.4-0.6 | 0.008 | 0.009 | 0.008 | 0.008 | 18 |
| > 0.6 | 0.000 | 0.001 | 0.000 | 0.000 | 24 |
| Mongu | | | | | |
| > 0.4 | 0.008 | 0.009 | 0.008 | 0.008 | 439 |
| 0.4-0.6 | 0.009 | 0.010 | 0.009 | 0.009 | 281 |
| > 0.6 | 0.006 | 0.008 | 0.007 | 0.007 | 158 |
| Mezaira | | | | | |
| > 0.4 | 0.002 | 0.002 | 0.002 | 0.002 | 1003 |
| 0.4-0.6 | 0.004 | -0.001 | -0.001 | -0.001 | 389 |
| > 0.6 | 0.000 | -0.002 | -0.002 | -0.002 | 614 |
| GSFC | | | | | |
| > 0.4 | 0.000 | 0.001 | 0.000 | 0.000 | 791 |
| 0.4-0.6 | -0.003 | -0.002 | -0.003 | -0.003 | 348 |
| > 0.6 | 0.003 | 0.003 | 0.002 | 0.002 | 443 |

Table 2. Average values, of the difference in the real part of refractive index (RPRI) retrievals from inversions using STD and REL assumptions on IPRI smoothness constraints. The difference is defined as REL-STD.






| AOD (440 nm) | VMR fine | STD fine | VMR coarse | STD coarse | Number of retrievals |
|---|---|---|---|---|---|
| Rexburg | | | | | |
| < 0.2 | 0.000 | -0.005 | 0.014 | 0.000 | 361 |
| > 0.2 | 0.000 | -0.004 | 0.011 | 0.002 | 84 |
| Mongu | | | | | |
| < 0.2 | 0.000 | -0.006 | 0.005 | 0.002 | 236 |
| > 0.2 | 0.000 | -0.002 | -0.020 | 0.000 | 660 |
| Mezaira | | | | | |
| < 0.2 | 0.000 | 0.000 | -0.002 | 0.000 | 696 |
| > 0.2 | 0.002 | 0.000 | 0.020 | 0.003 | 2487 |
| GSFC | | | | | |
| < 0.2 | -0.002 | 0.005 | -0.040 | 0.008 | 22542 |
| > 0.2 | 0.000 | -0.003 | -0.040 | 0.004 | 4320 |

Table 3. Average values, of the difference in in volume median radius (VMR) in microns and width of particle size distribution (STD) retrievals from inversions using STD and REL assumptions on IPRI smoothness constraints. The
difference is defined as REL-STD.







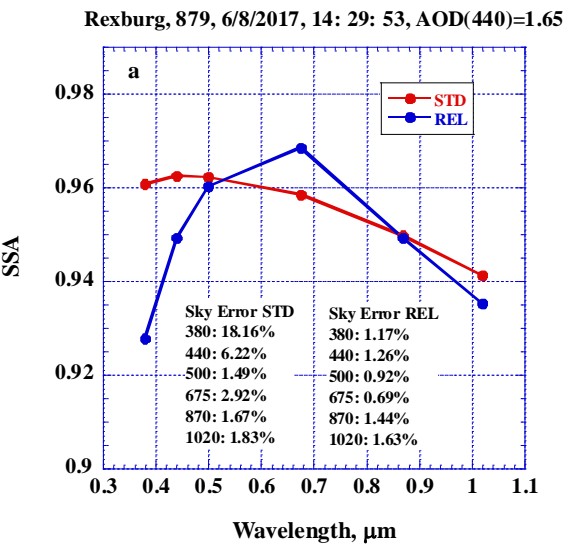
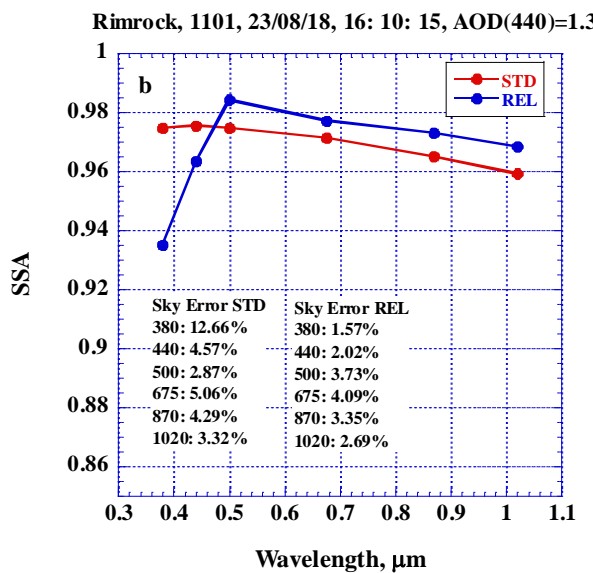

Figure 1. SSA retrieved using V3 assumptions on the IPRI smoothness constants (STD) and that of relaxed (REL): a) Rexburg (AOD(440)=1.65, AE=1.74) and b) Rimrock (AOD(440)=1.33, AE=1.79) AERONET sites.









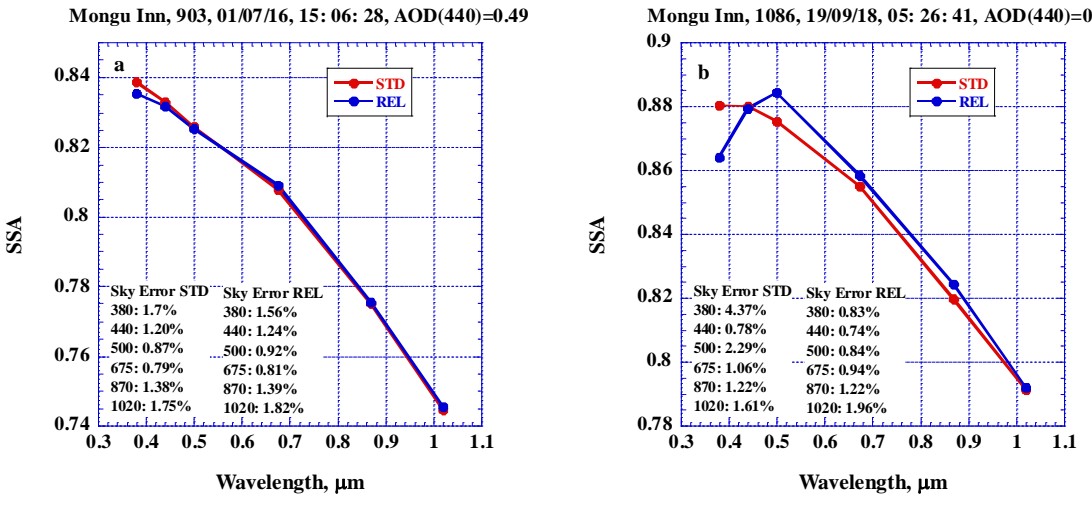

Figure 2. SSA retrieved using V3 assumptions on the IPRI smoothness constants (STD) and that of relaxed (REL) at Mongu Inn AERONET site: a) AOD(440)=0.49, AE=1.85 and b)AOD(440)=0.89, AE=1.97.









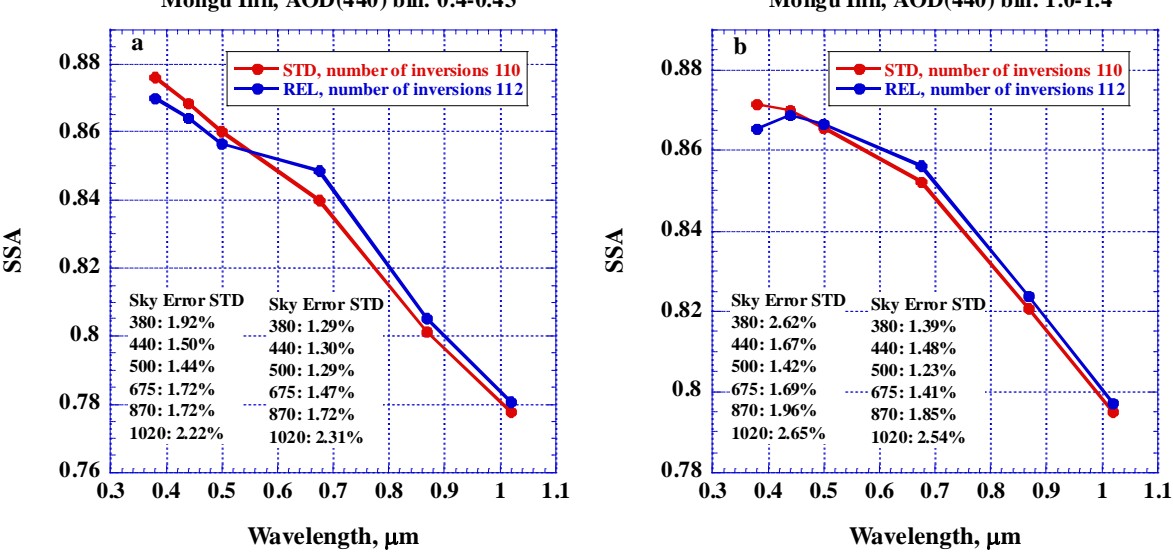

Figure 3. SSA averaged over AOD(440) bins for Mongu Inn AERONET site: a) 0.4-0.43 and b) 1.0-1.4









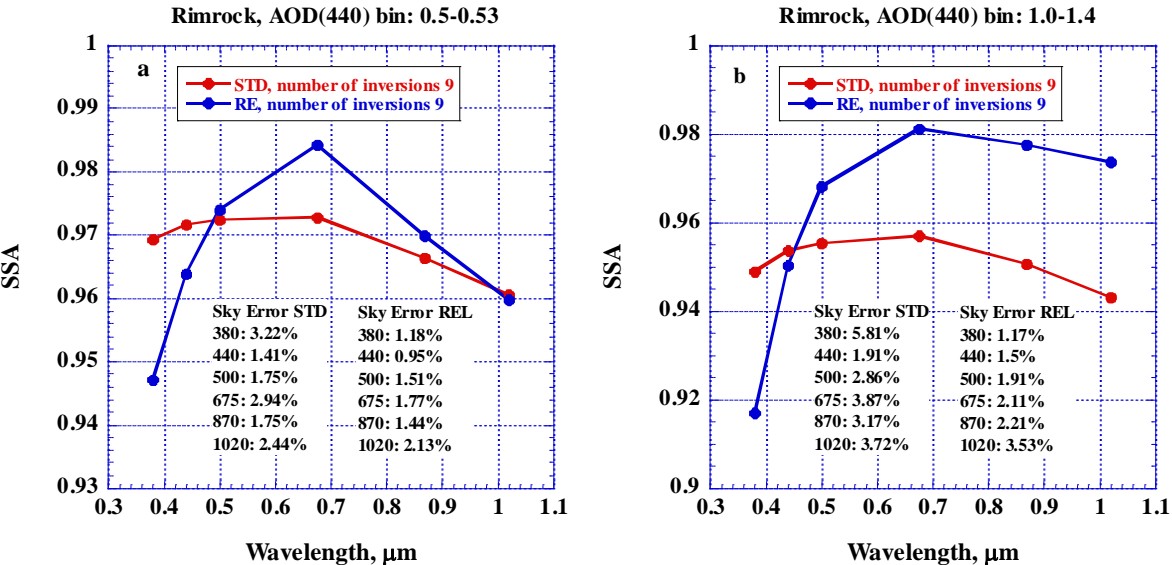

Figure 4. SSA averaged over AOD(440) bins for Rimrock AERONET site: a) 0.5-0.53 and b) 1.0-1.4






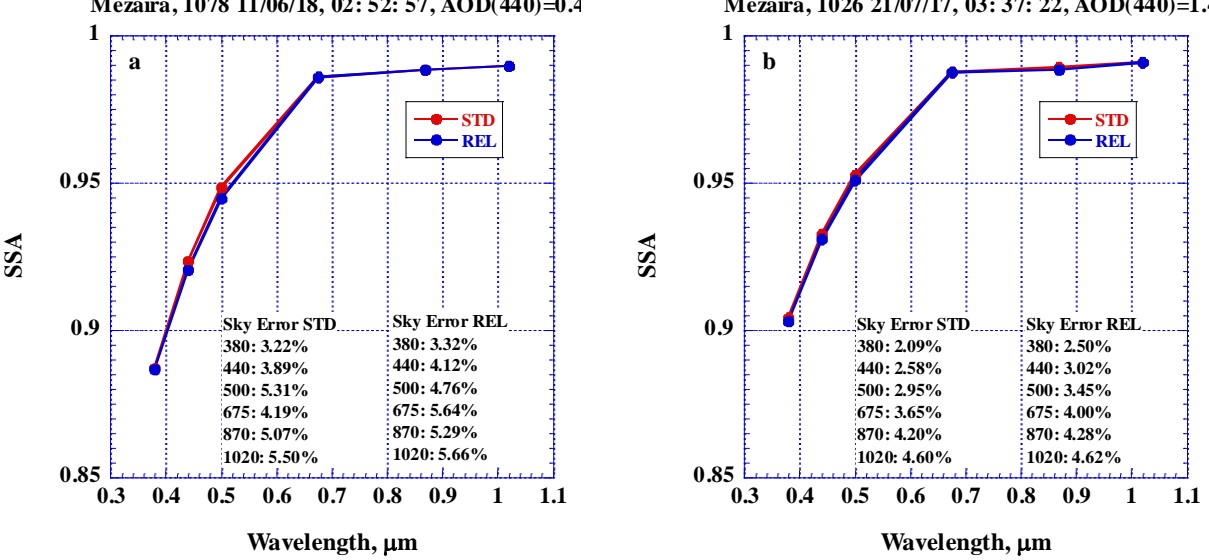

Figure 5. SSA retrieved using  V3 assumptions on the  IPRI smoothness constants (STD) and that of relaxed (REL) at Mezaira  AERONET site: a) AOD(440)=0.44, AE=0.24 and b) AOD(440)=1.45, AE=0.2.









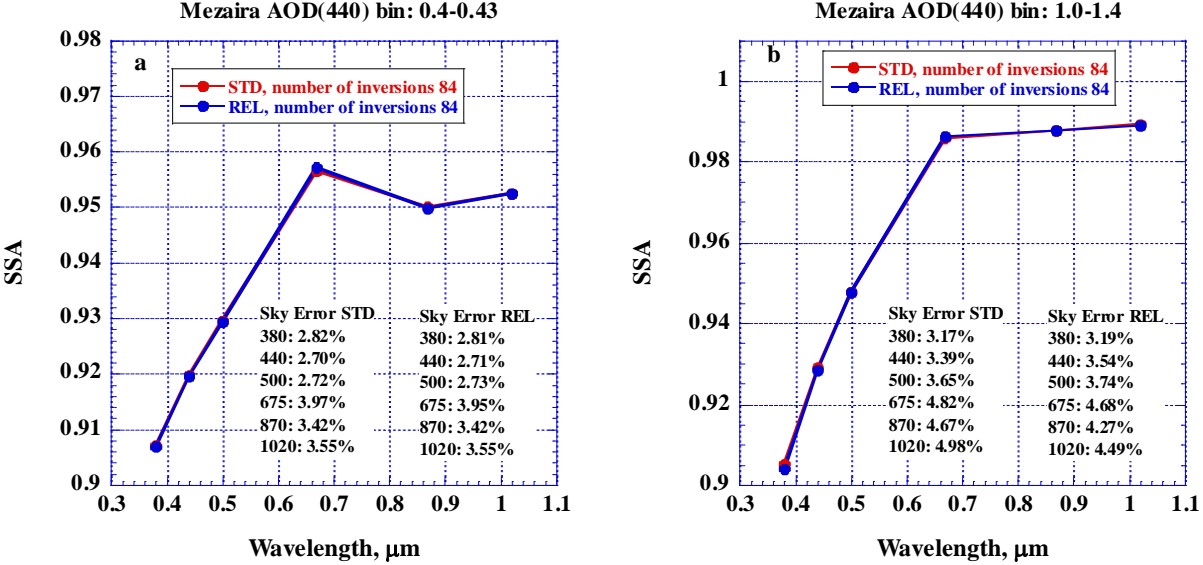

Figure 6. SSA averaged over AOD(440) bins for Mezaira AERONET site: a) 0.4-0.43 and b) 1.0-1.4.







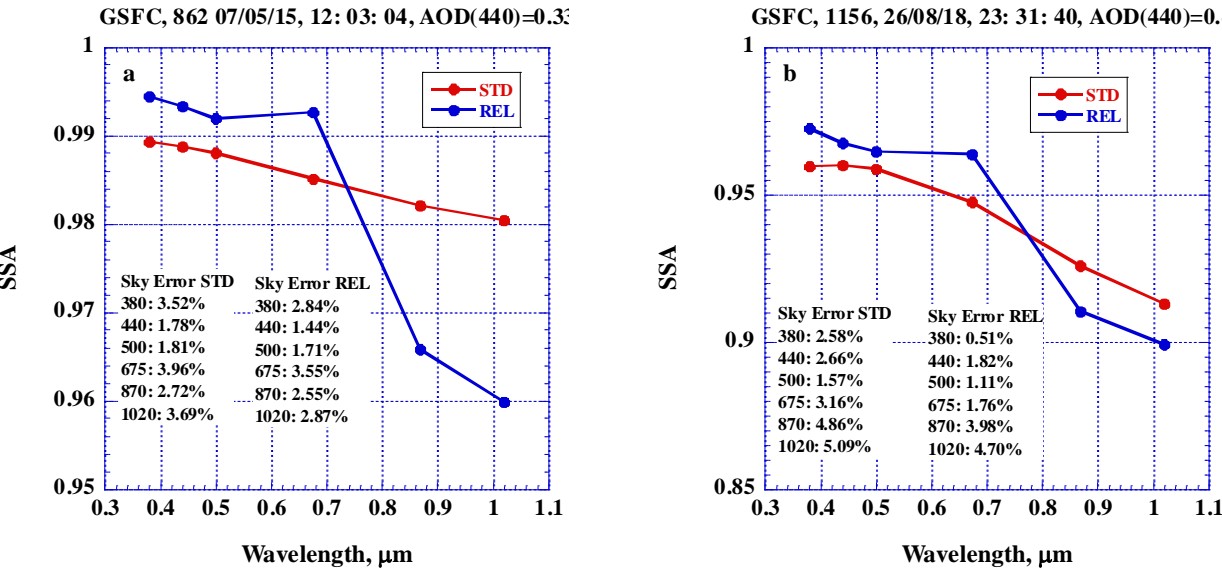

Figure 7. SSA retrieved using V3 assumptions, depicted as STD, and that of REL on the IPRI smoothness constants at GSFC AERONET site: a) AOD(440)=0.33, AE=1.74 and b) AOD(440)=0.49, AE=1.63.






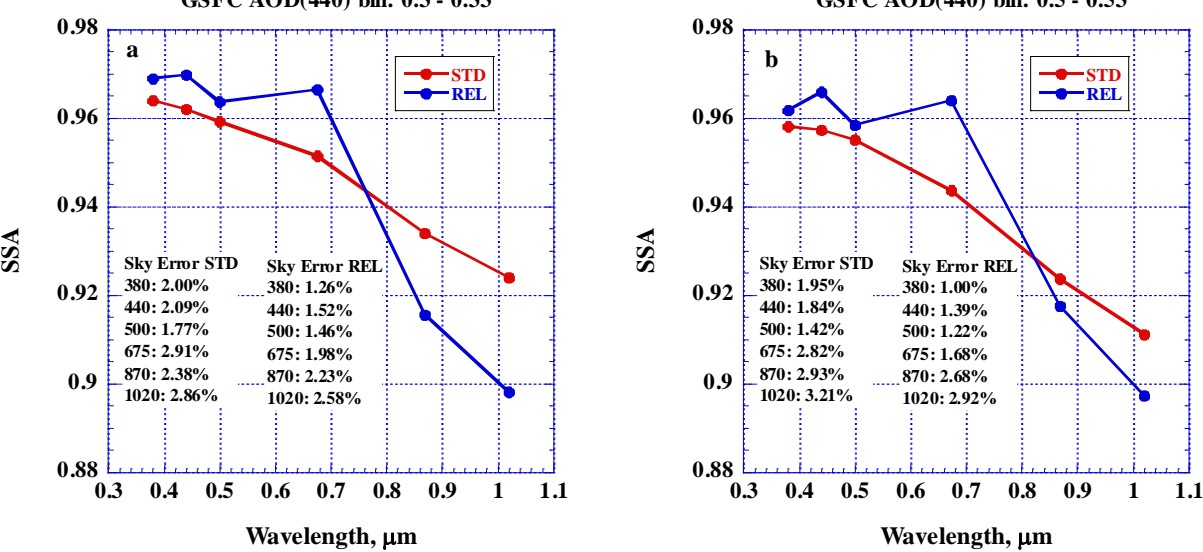

Figure 8. SSA averaged over AOD(440) bins for GSFC AERONET site: a) 0.3-0.33 and b) 0.5-0.53.









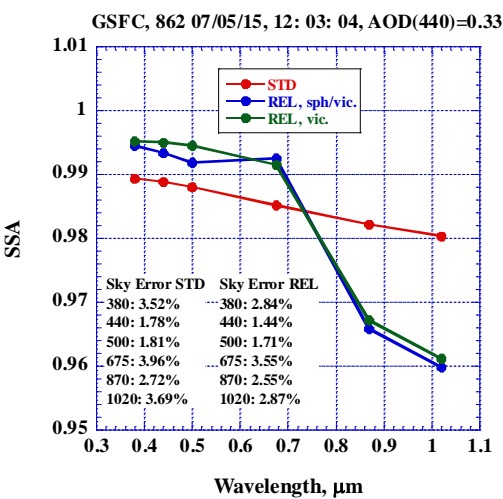

Figure 9. Same as Figure 7a with addition of SSA retrieved from vicariously calibrated sky radiances, depicted by REL, vic.
Abbreviation REL, sph./vic. refers to SSA retrieved by inverting sky radiances calibrated using integrated sphere at all the
channels except 380 nm which was calibrated using vicarious method.








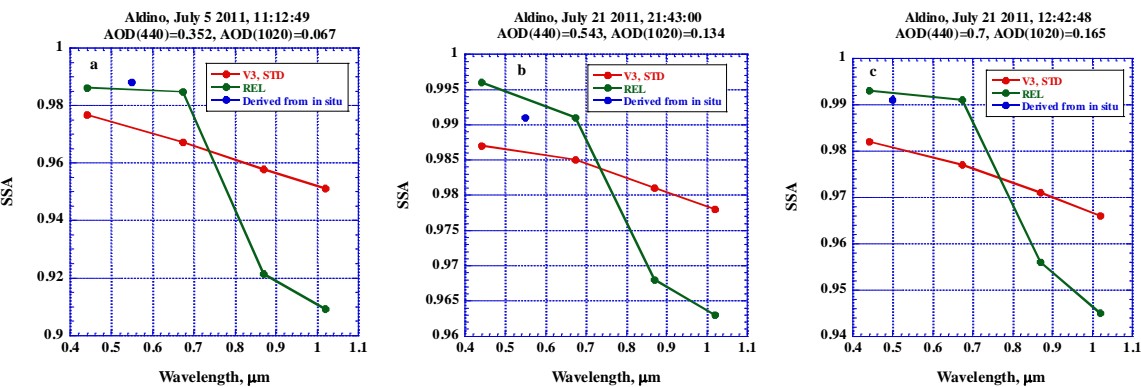


Figure 10. Comparison of SSA retrieved by V3 aerosol inversion code employing STD assumption on IPRI smoothness constraints and SSA retrieved by inversion code using REL assumption to SSA derived from in situ measurements at Aldino AERONET site: a) 440 nm AOD=0.352, b) 440nm AOD=0.543, c) 440 nm AOD=0.7.










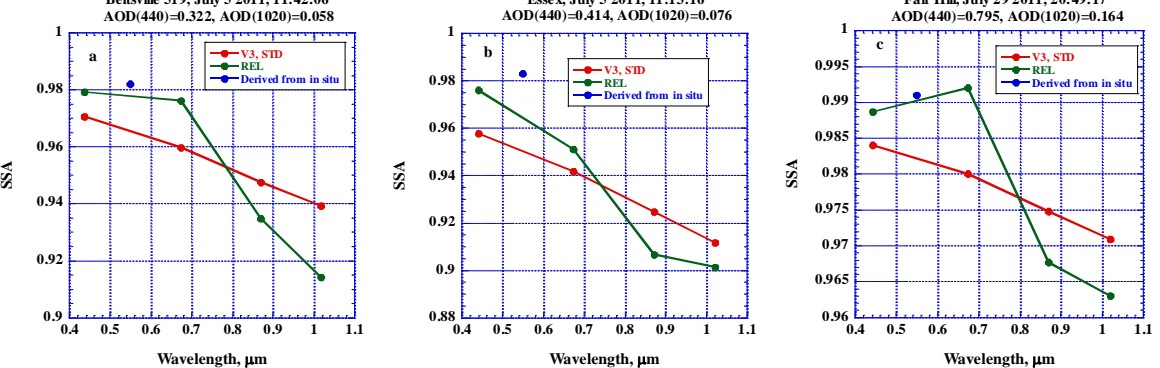

Figure 11. Comparison of SSA retrieved by V3 aerosol inversion code employing STD assumption on IPRI smoothness constraints and SSA retrieved by inversion code using REL assumption to SSA derived from in situ measurements: a) Beltsville AERONET site, 440 nm AOD=0.322, b)Essex AERONET site, 440nm AOD=0.414, c) Fair Hill AERONET site, 860   440 nm AOD=0.795.

