# Peer review of "Employing relaxed smoothness constraints on imaginary part of refractive index in AERONET aerosol retrieval algorithm."

_Atmospheric Measurement Techniques, 2022_

## Author Comment (AC1)

Reply to comments of Referee #3.

The authors would like to thank Referee #1 for careful reading of the manuscript and valuable comments.

**Questions:**

1. - What is the dominant aerosol component in Rexburg, Mongu, Mezaira, and GSFC, and how is it justified? Any supporting information to describe the air condition (particularly the aerosol composition) for selected cases. For example, readers do not know if BrC is really dominant for the cases treated in Fig. 1.

Unfortunately, detailed information on aerosol composition at selected sites is rarely exits since it would require many instrument types co-located for a full optical and chemical characterization of the aerosol. Nevertheless, we tried our best to add more information and also included three more references. One is for Mongu aerosol composition:

Eck, T.F., Holben, B.N., Ward, D.E., Mukelabai, M.M., Dubovik, O., Smirnov, A., Schafer, J.S., Hsu, N.C., Piketh, S.J., Queface, A. and Roux, J.L., 2003. Variability of biomass burning aerosol optical characteristics in southern Africa during the SAFARI 2000 dry season campaign and a comparison of single scattering albedo estimates from radiometric measurements. *Journal of Geophysical Research: Atmospheres*, *108*(D13).

Two others are related to Mezaira site which is dominated by desert dust: Reid, J.S., Reid, E.A., Walker, A., Piketh, S., Cliff, S., Al Mandoos, A., Tsay, S.C. and Eck, T.F., 2008. Dynamics of southwest Asian dust particle size characteristics with implications for global dust research. *Journal of Geophysical Research: Atmospheres*, *113*(D14),

Eck, T.F., Holben, B.N., Reid, J.S., Sinyuk, A., Dubovik, O., Smirnov, A., Giles, D., O'Neill, N.T., Tsay, S.C., Ji, Q. and Al Mandoos, A., 2008. Spatial and temporal variability of column-integrated aerosol optical properties in the southern Arabian Gulf and United Arab Emirates in summer. *Journal of Geophysical Research: Atmospheres*, *113*(D1).

Also, a sentence was added to clarify aerosol types for Rexburg and Rimrock: High aerosol optical depth events dominated by fine mode particles (high AE) in the US northern Rocky Mountain region in June-October are dominated by biomass burning emissions.

2. Can we generalize the finding and lessons in this study based on this four sites only?

The main idea of the manuscript is that no strong smoothness constraints are imposed on spectral dependence of the imaginary part of refractive index of aerosols of any type for the new REL retrievals. Therefore, there is no artificial suppression of spectral dependence of aerosol absorption. We think this is general statement applicable to all aerosol types including mixtures of aerosols of different types. The analysis for four sites dominated by four main aerosol types (BC, BrC, dust, industrial) demonstrated successful performance of inversions employing the new relaxed smoothness constraints over a wide range of aerosol characteristics .

3. Why this new REL only make some change for the BrC-dominated biomass burning (BB) aerosols, not the mineral dust and BC-dominated BB aerosols, which are other radiative absorbing aerosols?

This is because in V3 the constraints on the spectral IPRI were already relaxed for coarse mode mineral dust. In V3 the Lagrange multiplier for spectral variation of the imaginary part was interpolated from dust

(low AE, low value of Lagrange multiplier, relaxed smoothness constraints) to fine mode (high AE, high value of Lagrange multiplier, strong smoothness constraints). For BC dominated aerosols strong smoothness constraints in V3 for the imaginary part of refractive index worked due to the flat spectral dependence of IPRI of BC. This is described in Introduction and Section 2.

4. This new REL can help the retrieval of qualified SSA data in 340 nm channel?

We think it possibly can, depending on the data quality. In AERONET, we started looking at 340 nm to estimate if the sky radiance signal is strong enough above the noise level and if we can accurately calibrate this channel for sky radiance measurement.

5. This new REL enable us to have SSA data more under the low AOD case; Usually the SSA analysis relates to the polluted case, at least AOD > 0.4 due to the uncertainty issue. It is curious to see if this REL can lower the uncertainty of SSA retrieval in less-polluted case. In other words, application of new REL is only helpful for the retrieval in the polluted condition (often related to the high AE because of the general contribution of find mode particle to the large air pollution), or it is also useful to improve the retrieval in the lower AOD case of polluted (urban) area where the brown carbon is dominant.

Employing REL does not increase sensitivity to aerosol absorption, therefore the AOD>0.4 condition still stands in order to have the same uncertainty in retrieved SSA. The REL constraint can lower the sky residual error in all the cases where STD were suppressing spectral dependence of the imaginary part of refractive index and therefore result in more retrievals that meet the L2 sky error criteria. The possibility of lowering the AOD threshold for SSA retrieved in polluted areas depends on the accuracy of the sky radiance calibration. Alternative calibration methods with higher accuracy will be investigated after enough statistics of REL inversions will be available.

Minor and specific comments:

1. - Nowadays, there are so many AERONET stations and really long-term measurement data have been accumulated. In this situation, it is curious if we really can apply the analysis result only based on a several cases to the general situation (only some days were selected for the analysis even in only 'four' sites). The analysis in this study looks qualified with reliable cases showing clear dominance of a target aerosol composition, which can be a representative example for the meaningful discussion for new REL impact under the certain situation. But still, it seem the limited discussion because now we have so abundant information of AERONET measurement for several hundreds of local stations. Thus, the statistical analysis using the large number of dataset will be more expected for the generalization of findings in this work. In my opinion, this manuscript can be a good paper as a case study to show the 'possibility' for the usage of new REL for better expression of BrC optical properties. But it may be better to prepare another manuscript for the 'generalization of finding in this study'. In the second manuscript, the statistical analysis looks much required.

We absolutely agreed. REL constraints will be the part of upcoming Version 4 of AERONET aerosol retrieval algorithm. All the data will be reprocessed with REL and statistics of the difference between STD (V3) and REL (V4) for all the sites will be produced, analyzed, and published.

Abstract seems too long, so the key point of this study is not well transferred to readers. The word number in this abstract is > 800, which looks too much compared to the general criteria (~ 200 to 300 words. I do not know the limitation of word number in ACP/AMT).

We do not find any abstract limitations for AMT publication. However, we reduced the abstract by  $\sim$ 20% which we think makes it more concise and clearer.

3. Line 46-50: Two sentences are not connected well (First one mentioned DRAGON campaign, but second on mentioned the DISCOVER-AQ campaign. How to connect the story here?)

**In new, shorter version of the abstract these sentences were removed.**

4. Line 56-57: How to understand this sentence? (What is the relationship between the insigficant impact for the mineral dust and relatively small impact for the BC-dominated biomass burning aerosol?)

**This sentence was removed in new version of abstract.**

5. Line 104-106: The reference or clues are required to raise this issue about the BrC. Now there is no surporting information associated with this statement, which looks very essential for the motivation of this study.

The following reference is added to this sentence: Kirchstetter, T., W., and Thatcher, T., L.: Contribution of organic carbon to wood smoke particulate matter absorption of solar radiation, Atmos. Chem. Phys., 12, 6067-6072, 2012.

6. - Line 214: => For example,

**Corrected.**

7. - Line 242: BrC carbon  $\Rightarrow$  BrC

**Corrected.**

8. Line 252-255: I am not sure if this kind of discussion is possible without any fire or humidity information in this case.

This is a discussion on the possible range of conditions and relative phase of combustion from biomass burning and the resulting variation in composition of the aerosols. It is pertinent to the differences in spectral SSA that are being discussed. References to support this discussion are also given.

9. - Line 270-307: I am not sure the existence of BrC in GSFC, but It may be also possible to see high amount of BrC in the urban region in the urban region, and the BrC pattern (e.g., hygroscopic growth related to the extent of aging) can be regionally different: (e.g., Zhang et al., GRL, 2011, https://doi.org/10.1029/2011GL049385). It will be useful to see if there is difference of the BrC optical pattern between the biomass burning and urban area in the further study.

We agreed. As soon as a large number of REL inversion will be available this will be investigated. Actually, separate and detailed future research on absorbing aerosols is planned.

10. - Line 360-363: This manuscript does not have the chapter of 'data description' or 'methodology'. So there is no information of SSA from in-situ measurement in DRAGON-MD campaign. A short phrase to mention Schafer et al. (2014) may not be enough because the SSA estimation using in-situ measurement itself can make the large difference from the optically measured SSA (e.g, surface representative vs. column information). so at least several statements about the data/methodology of in-situ SSA calculation looks needed.

The following modification was done including additional sentences:

The SSA values are derived from in situ measurements made during aircraft vertical profiles of scattering and absorption coefficients at 550 nm. For each profile, 1 s sampled values of scattering coefficient measurements at 450, 550, and 700 nm from the nephelometer and absorption coefficient measurements at 470, 532, and 660 nm from the Particle Soot/Absorption Photometer were provided, both from dried air samples. At 550 nm, an additional scattering measurement at ambient relative humidity allowed for the calculation of an ambient SSA (rather than dried aerosol) that is more suitable for comparison with the SSAs derived from AERONET radiance measurements. In order to produce a column SSA value to compare with AERONET, the 1 s SSA aircraft measurements were averaged for the duration of the profile sampling after weighting the values according to aerosol loading (Schafer et al., 2014).

---

## Author Comment (AC2)

Reply to comments of Referee #1.

The authors would like to thank Referee #1 for careful reading of the manuscript and valuable comments.

**Comments:**

1. The abstract is too long and difficult to understand.

We reduced abstract by  $\sim 20\%$  which we think makes it more concise and clearer.

2. The paper is overloaded with abbreviations. Many readers who are interested in details of AERONET retrieval algorithm will want just quickly browse the paper. In this case, the text is often too difficult to understand due to abbreviations that are not very common.

For easy reference, we summarized the list of abbreviations in Table A1. At the same time all the abbreviations were removed from abstract to make it easier to read without need to look for meaning of each abbreviation. I addition, we changed some abbreviations to more commonly used in aerosol community: ASD was replaced by PSD.

**3.** Row 77. "Absorption at 380 nm is particularly important as this is the wavelength range that satellite observations and algorithms are able to retrieve atmospheric column absorption from existing (Jethva et al, 2014) and future satellite sensors (Werdell at al., 2019)". This statement should be clarified. Is it due to low aerosol optical depths at longer wavelengths?

The following explanation was added to the manuscript: The unique utility of measurements in UV for satellite remote sensing is related to increased sensitivity to aerosol absorption due to absorption of molecular scattering by aerosols (e. g. Torres et al., 1998)

4. Equation (3) is discussed too briefly, it is better to say a few words about what each term in it represents, rather than just defining the variable.

The following sentence was added: Jacobian matrices are the matrices of the first derivatives of measurements with respect to retrieved parameters and covariance matrices are diagonal matrices with elements equal to the accuracy (variances) of the measurements and/or a priori estimates.

5. There are two  $\gamma$  variables in the equation (3), Lagrange multiplier  $\gamma_n$  (row 149) and Lagrange multiplier  $\gamma_k$  (row 146). Later in the text the authors discuss the Lagrange multiplier  $\gamma_3$  (row 164). It is necessary to write explicitly which one Lagrange multiplier is considered.

To avoid confusion the following changes in equation (3) have been made: the separated symbols were introduced for Lagrange multipliers contributing to optical measurements ( $\gamma_k^m$ ) and to smoothness constraints ( $\gamma_n^s$ ). In this case  $\gamma_3^s$  corresponds to smoothness constraints on the imaginary part of refractive index.

6. The authors discuss AERONET measurements from many observational places scattered throughout the world. A table with geographical coordinates of these places would be very helpful. Please check if the name Mesaira is spelled correctly (Mesairaa?).

Geographical coordinates are added. Mezaira is spelled correctly.

7. The authors analyze the improvement achieved with use of the 'new' smoothness constraints by comparing the wavelength dependence of the retrieved aerosol single scattering albedo (SSA) using the 'old' and 'new' version of the constraints. They consider the dependence of SSA on wavelength for different aerosol optical depth (AOD) bins. Fig. 1 shows SSA wavelength dependences for Rexburg and Rimrock observational places. Later, wavelength dependences of SSA are presented for AOD bins only for Rimrock (4), but not for Rexburg. Tables 1-3 present further analysis only for Rexburg, but not for Rimrock. The analysis should be done in uniform manner.

The reasons of separating Rexburg and Rimrock data are the following. Rimrock AOD varies in wider range than that of Rexburg. Therefore, for plots presenting SSA retrievals averaged over AOD bins Rimrock was selected due to better sampling than in case of Rexburg statistics of SSA retrievals for both moderate (0.5-0.53) and high (1.0-1.4) AOD bins. For Rexburg there are just a few SSA retrievals corresponding to high AOD measurements. From other hand, overall statistics of aerosol retrievals for Rexburg is better than for Rimrock (it is just 18 retrievals total). That is why Rexburg was selected for comparison of STD-REL retrievals. In addition, the aerosol type which dominated the loading over Rexburg and Rimrock are similar: smoke from regional forest fires.

8. It is not clear how many observations are used to produce figures 1-11.

There are two types of figures: individual cases (like Fig., 1) and retrievals averaged for AOD bins (like Fig. 4). For Figures of the second type the number of retrievals used in averaging is presented on figures.

9. Presence of error bars on the plots would simplify its understanding.

Error bars in Version 3AERONET aerosol product are calculated for four standard channel retrievals. For the expanded set of wavelengths this is not currently done. They will be available in forthcoming Version 4.

10. Tables 1-3 show absolute differences between aerosol parameters retrieved using 'old' and 'new' constraints. I suggest including relative differences also because in many cases the absolute differences are too small and seeing so many zeros in the tables is not very informative.

In fact, absolute differences are what is normally used to characterize uncertainty of aerosol retrievals. For example, in SSA case the 0.03 threshold is often used to determine the suitability

of SSA retrievals for use in radiative forcing simulations. Relative differences are not so clearly interpreted. However, to facilitate the comparison, standard deviation was added to Tables 1-3.

11. The analysis presented in Tables 1-3 was done for wavelengths 440,675,870,1020 nm. Why is the 380 nm wavelength excluded from the analysis?

Analysis was done for the four standard channel inversions to estimate the effect of using REL smoothness constraints on aerosol retrievals in Version 3. The 380 nm wavelength is not a part of AERONET standard inversion product yet.